# Research Progresses and Applications of Fluorescent Protein Antibodies: A Review Focusing on Nanobodies

**DOI:** 10.3390/ijms24054307

**Published:** 2023-02-21

**Authors:** Yu-Lei Chen, Xin-Xin Xie, Ning Zhong, Le-Chang Sun, Duanquan Lin, Ling-Jing Zhang, Ling Weng, Tengchuan Jin, Min-Jie Cao

**Affiliations:** 1College of Ocean Food and Biological Engineering, Jimei University, Xiamen 361021, China; 2Institute of Health and Medicine, Hefei Comprehensive National Science Center, CAS Key Laboratory of Innate Immunity and Chronic Disease, Division of Life Sciences and Medicine, University of Science & Technology of China, Hefei 230007, China

**Keywords:** fluorescent protein, monoclonal antibody, nanobody, research progress, application

## Abstract

Since the discovery of fluorescent proteins (FPs), their rich fluorescence spectra and photochemical properties have promoted widespread biological research applications. FPs can be classified into green fluorescent protein (GFP) and its derivates, red fluorescent protein (RFP) and its derivates, and near-infrared FPs. With the continuous development of FPs, antibodies targeting FPs have emerged. The antibody, a class of immunoglobulin, is the main component of humoral immunity that explicitly recognizes and binds antigens. Monoclonal antibody, originating from a single B cell, has been widely applied in immunoassay, in vitro diagnostics, and drug development. The nanobody is a new type of antibody entirely composed of the variable domain of a heavy-chain antibody. Compared with conventional antibodies, these small and stable nanobodies can be expressed and functional in living cells. In addition, they can easily access grooves, seams, or hidden antigenic epitopes on the surface of the target. This review provides an overview of various FPs, the research progress of their antibodies, particularly nanobodies, and advanced applications of nanobodies targeting FPs. This review will be helpful for further research on nanobodies targeting FPs, making FPs more valuable in biological research.

## 1. Introduction

Fluorescent protein (FP) is the most commonly used tool protein in biomedical science research. Following the increased use of FP, FP antibodies have also been developed. Compared to the traditional antibody (Figure 1A), heavy-chain antibodies (HCAb) from camelid (Figure 1B) and immunoglobulin new antigen receptors (IgNAR) from sharks (Figure 1C) naturally lack light chains. Single-domain antibody (sdAb) is an antibody entirely composed of variable domains of HCAb or IgNAR. Its molecular weight is 12–15 kDa, which is about 1/10 the size of a monoclonal antibody; hence, it is also referred to as nanobody. The nanobody, with a smaller size, high stability against acid and heat, strong penetration capability, ease of genetic engineering and expression, and high affinity, has been applied in various fields by scientific researchers. It can easily access grooves, seams, or hidden antigenic epitopes on the surface of the target, recognizing many antigens that conventional antibodies cannot recognize [1]. Hence, nanobodies have been used as biomolecules in organisms and cell culture systems in developmental biology, crystalline chaperones for the conformational state in stable biology, regulators or inhibitors of enzyme activity, immunohistochemical reagents for biochemical analysis, and secondary antibodies in immunoblotting and immunofluorescence [2,3,4,5,6]. Nanobodies that specifically bind FP are widely used in subcellular localization, intracellular signaling pathway studies, live cell imaging, and targeted nanomaterials [7,8,9]. In particular, nanobodies have unique advantages over traditional IgG antibodies in super high-resolution imaging.

## 2. Overview of Fluorescent Proteins

### 2.1. The Discovery of Fluorescent Proteins

Luminescence is a common phenomenon in marine invertebrates. Coelenterates, including jellyfish, hydra, and coral, emit green fluorescence under ultraviolet (UV) or blue light, while ctenophores emit blue fluorescence. Green fluorescent protein (GFP) was first identified in *Aequorea victoria* by Shimomura et al. in 1962 [10], and then cloned and expressed in 1985 [11]. The unique properties and stable structure of wild-type GFP (wtGFP) have, accordingly, attracted many researchers to study it. Since then, based on the existing knowledge and its crystal structure, Tisen and his colleagues generated various GFP proteins with different fluorescent properties by mutation and elaborated the luminescence mechanism of GFP [12]. Concerning GFP-related research, three scientists (Shimomura, Chalfie, and Tsien) won the 2008 Nobel Prize in Chemistry for discovering GFP and their outstanding contributions to its application.

Ancestral red fluorescent protein (RFP), namely, DsRed, was cloned from non-bioluminescent reef corals in 1999 [13]. RFP can be used together with GFP to solve scientific problems that GFP alone cannot solve. Most importantly, due to the low background of RFP in intracellular imaging, it is more suitable for applications in bioscience research [14,15]. With the efforts of many scientists, the fluorescence spectra of FPs have been reported to cover the entire visible region and even the near-infrared region, providing a wealth of tools for visualizing and quantifying proteins in living cells [16]. Figure 2 summarizes the development of FPs.

### 2.2. Structure and Functional Properties of GFP

GFP is a naturally occurring, globular, and soluble acidic FP. The primary structure of GFP consists of a monomer composed of 238 amino acid residues, with a molecular weight of 27–30 kDa. Its crystal structure displays a center formed by 11 anti-parallel β chains, forming a cylindrical, tightly packed β-barrel structure. The β-barrel center is protected by a light-emitting group of 4-(p-hydroxybenzylidene)-5-imidazolinone connected to its α-helix, with both ends sealed by short α-helix segments [17,18]. The maturation of the chromophore requires no cofactor other than O_2_ [19]. GFP absorbs blue light or UV light and emits green fluorescence, with the main excitation peak at 395 nm, the lowest excitation peak at 475 nm, and the emission peak at 509 nm. GFP is extremely stable and can easily withstand high-temperature treatment. Formaldehyde fixation or paraffin embedding does not affect its fluorescence characteristics. Through the in-depth investigation of the structure and maturation process of FPs, more and more researchers redesign the structure of FPs to provide them new functions and properties, further promoting their development and application.

### 2.3. GFP Derivatives

The limiting factors in GFP applications are pH, chloride ion sensitivity, and poor photostability. Targeted modifications of GFP correct these flaws, boosting its quantum yield, brightness, molar extinction coefficient, and photostability. In addition, several GFP derivatives have been established to broaden the range of colors emitted, including blue, ultramarine, cyan, and yellow.

In response to the low fluorescence intensity of wtGFP under blue light excitation and its unstable expression in mammalian cells, Zhang et al. engineered an amino acid substitution of GFP (F64L and S65T), namely, enhanced GFP (EGFP), whose fluorescence intensity is increased by 35-fold [20]. EGFP remains the most commonly used FP mutant in the green light region, given its good photostability and high brightness. In addition, the folding properties of GFP have been optimized, obtaining a superfolder GFP (sfGFP) with super folding ability. The sfGFP has six new mutations, including S30R, Y39N, N105T, Y145F, I171V, and A206V. sfGFP can efficiently fold even when expressed in fusion with insoluble proteins, ensuring the brightness of its fluorescence [21]. GFPuv, another modified form of GFP, is optimized for fluorescence under UV light. It generates brighter green fluorescence in the presence of UV light with an excitation peak at 395 nm and an emission peak at 509 nm and can be utilized to pinpoint the location of different intracellular proteins [22,23]. Three amino acid substitutions occur in GFPuv, including F99S, M153T, and V163A. These mutations induce an 18-fold stronger expression of GFPuv than wtGFP in *E. coli*. The *GFPuv* gene is a synthetic gene where the five low-frequency Arg codons are replaced by codons suitable for *E. coli* to ensure effective expression of GFPuv.

The site-directed mutation of Y66H constructs the blue fluorescent protein (BFP) with an excitation peak at 384 nm and an emission peak at ~445 nm. Its excitation spectrum is close to UV light, which damages cells during operation, and its short emission wavelength induces autofluorescence in cells. Enhanced BFP (EBFP) is weakly luminescent and poorly resistant to photobleaching and acid, with a high background signal for cellular imaging. Three brighter EBFP mutants have been developed, including Azurite [24], EBFP2 [25], and mTagBFP [26], which are 1.6-, 2.0-, and 3.7-fold brighter than EBFP, respectively. mTagBFP, the brightest BFP, is derived from a mutant of the red fluorescent protein (RFP) TagRFP [27], with an extinction coefficient of 52,000 M^−1^ cm^−1^ and a quantum yield of 0.63, which significantly improves its resistance to photobleaching. Mutation of Y66F in wtGFP results in an ultramarine fluorescent protein with an emission wavelength of 442 nm [28]. However, the low fluorescence quantum yield of the GFP-Y66F variant severely limits its applicability in imaging. Further mutation of GFP-Y66F with amino acid substitutions of T65Q, Y145G, H148S, and T203V results in the ultramarine fluorescent protein Sirius, which is 25 times brighter than GFP-Y66F [29].

Cyan fluorescent protein (CFP) has an excitation peak at 449 nm and an emission peak at ~482 nm, with spectral properties ranging between BFP and EGFP. Similarly to BFP, a site-directed mutation of Y66W is used to construct CFP. CFP has wide applications in multicolor imaging and localization research due to its outstanding photostability. For example, the gene encoding CFP has been optimized for human codon preference and is now commercially available under the trade name AmCyan1 (Clontech). Additionally, there are several mutants brighter than enhanced CFP (ECFP), such as Cerulean, mTFP (twice as bright as Cerulean), and mTurquoise (about twice as bright as ECFP) [21,30].

Mutations in the yellow fluorescent protein (YFP) are not restricted in the core motif of the chromophore Y66 but in residues structurally adjacent to Y66. The excitation peak of YFP is ~518 nm, with an emission peak of ~531 nm. Compared with GFP, the fluorescence of YFP shifts toward the red spectrum, which is mainly due to the substitution of Thr with Tyr at position 203. Enhanced YFP (EYFP) is one of the most fluorescent and extensively utilized FP biosensors to detect intracellular pH and chloride ion concentrations. However, it is very susceptible to acid and chloride ions and is less photostable than many jellyfish-derived FPs [31]. mCitrine and mVenus, improved versions of EYFP, are currently the most used YFP [32,33]. The photophysical properties of GFP and its derivatives are summarized in Table 1.

### 2.4. RFP Derivatives

RFP is another widely used FP. DsRed is a tetramer that tends to form multimers when performing protein fusions and can be toxic when expressed intracellularly, so it has been engineered to obtain monomers. The most notable RFP is the “mFruit” family, including mCherry, mBanana, mOrange, dTomato, mTangerine, and mStrawberry (Table 1), of which mCherry, with mutations of K163Q and K83L, is the most preferred. mCherry has fast maturation, good monomeric properties, and better photostability, although it is less bright [34]. However, TagRFP in its monomer form is about three to four times brighter than mCherry, making it a relatively bright monomeric RFP that is currently available [27]. mKate is a novel monomeric far-red fluorescent protein derived from a four-site mutation of TagRFP with an excitation peak at 588 nm and an emission peak at 635 nm, which is only 45% as bright as EGFP; its dimerization protein, Katushka, is 67% as bright as EGFP [35]. mKate2, a mutant of mKate with amino acid substitutions of V38A, S165A, and K238R, has approximately double the brightness of mKate [36]. Compared to GFP and RFP, mKate and Katushka exhibit better imaging depth and richer optical signals for intra-object fluorescence imaging [35]. Hence, developing far-red fluorescent proteins is more valuable for in vivo bioimaging.

### 2.5. Near-Infrared FPs

Near-infrared (NIR) FPs are highly desired as protein tags in imaging applications. Most NIR FPs are designed from bacterial phytochrome photoreceptors (BphPs) [37]. The first NIR FP used in live animal imaging is IFP1.4, a fluorescent protein derived from BphP that fluoresces by binding to a biliverdin (BV) chromophore [38]. By DNA shuffling and random mutagenesis, a brighter IFP2.0 is then developed [39]. Although monomeric IFP1.4 and IFP2.0 can be achieved by breaking the dimerization interface of BphP, they still tend to dimerize at high concentrations. A natural monomeric infrared fluorescent protein (IFP), mIFP, was designed in 2015 [40]. mIFP has been proven to have sound imaging effects in *Drosophila* larvae and neurons. Shcherbakova et al. designed three bright monomeric NIR FPs with distinct spectra, namely, miRFP670, miRFP703, and miRFP709 [41]. The BphP-derived NIR FPs minimally require two domains, PAS (Per-ARNT-Sim) and GAF (cGMP phosphodiesterase-adenylate cyclase-FhlA), to covalently attach a BV chromophore and also possess a complex “figure-of-eight knot” structure topologically linking the GAF and PAS domains, which affects their folding. Hence, another class of bacterial photoreceptors, allophycocyanins (APCs), is used to engineer NIR FPs, such as smURFP [42]. Although APC-based NIR FPs are smaller, they bind BV less efficiently, leading to significantly lower brightness in mammalian cells than BphP-derived NIR FPs. To overcome the defects of BphP- and APC-based NIR FPs, a single-domain NIR FP named miRFP670nano was developed from cyanobacteriochrome (CBCR), representing the first CBCR-derived NIR FP that can efficiently bind endogenous BV chromophore and emit bright fluorescence in mammalian cells [43]. An essential advantage of miRFP670nano over BphP-derived NIR FPs is high photostability. The enhanced miRFP670nano3, with 14 mutations relative to the parental miRFP670nano, exhibits similar photostability to miRFP670nano [44]. Table 2 summarizes the photophysical properties of NIR FPs.

The discovery and application of FPs have provided a powerful research tool for modern biology. Nowadays, FPs have become one of the common tools scientists use to extend their applications to many research areas, such as gene expression regulation, organelle labeling, signal transmission, drug screening, and biomolecular interactions. Most cloning vectors expressing FPs have been commercialized and are available through commercial companies.

## 3. Research Progresses on Fluorescent Protein Antibody

### 3.1. Monoclonal Antibody Targeting Fluorescent Proteins

Antibody is a kind of immunoglobulin secreted by B lymphocytes. Monoclonal antibodies consist of two heavy chains and two light chains, with the heavy chain consisting of one variable region (VH) and three constant regions (CH), and the light chain consisting of one variable region (VL) and one constant region (CL). The heavy chains are covalently linked by disulfide bonds, and the CL region of the light chain is non-covalently linked to the CH1 domain of the heavy chain to form a stable antibody molecule [45]. Due to their specific receptor binding capacity, monoclonal antibodies produce a variety of biological activities, such as classical blocking, neutralization, complement activation, the killing of target cells through the Fc receptor, and the regulation of immune activity. They are critical biological macromolecules widely applied in immunoassay, in vitro diagnostics, and drug development. Polyclonal antibodies are a mixture of heterotypic antibodies derived from the immune response process of multiple B cells, and each antibody recognizes a different epitope of the same antigen [45]. Compared to polyclonal antibodies, monoclonal antibodies specifically detect an epitope on the antigen and are less likely to cross-react with other proteins, thus producing low background staining signals. Moreover, the reproducibility of results concerning monoclonal antibodies is higher than that of polyclonal antibodies under the same experimental conditions.

GFP is a unique in vivo reporter that can be analyzed for gene expression in many species. Gengyoando and Mitani used the glutathione-S-transferase (GST) fusion protein, which contains the full-length GFP coding region and a synthetic peptide corresponding to residues Ser208-His217 of GFP, as an immunogen to create the monoclonal antibody 65B12 against GFP [46]. Immunoblot analysis demonstrates that 65B12 specifically bound the GFP fusion protein. Furthermore, it can recognize the fluorescent cells in transgenic animals expressing the uric-86-gfp reporter construct; hence, it can be applied in immunohistochemistry. Zhuang et al. developed an improved method to purify GFP protein [47]. The monoclonal antibody against GFP, FMU-GFP.5, was prepared by immunizing mice using purified GFP as an antigen. GFP with high purity (>97% homogeneity) and sample yield (>90%) is purified using a straightforward 2-step technique using mAb FMU-GFP.5-coupled Sepharose 4B resin. In addition, all the functional recombinant target proteins coupled to GFP can be easily and directly isolated from cells, owing to the GFP epitope. These data suggest that this method is more effective in purifying GFP than any available method and resolves the low yield and purity challenge in most GFP purification methods.

### 3.2. Nanobody Targeting Fluorescent Proteins

#### 3.2.1. Introduction of Nanobody

The inherent properties of monoclonal antibodies, such as their large molecular weight, complex structure, and limited biological activity, have increasingly restricted their further applications. Therefore, it is urgent to develop alternatives to monoclonal antibodies. In 1989, Ward et al. prepared a variable domain of heavy-chain only antibody with weak binding ability toward lysozyme, spiking interest in sdAb research [48]. In 1993, Hamers-Casterman et al. discovered a new type of antibody in camel serum for the first time, which is entirely different from the traditional mammalian antibody [49]. This type of antibody is named HCAb due to its natural lack of light chains. HCAb consists of CH2, CH3, the hinge region, and the variable domain of HCAb (VHH), but it still has full antigen-binding capacity. In 1995, Greenberg et al. discovered heavy-chain only antibodies, known as IgNAR, in nurse shark [50]. It exists in both secretory and membrane-bound forms, and consists of a variable region, named variable new antigen receptor (VNAR), and several constant regions. Subsequently, IgNAR has been found in wobbegong shark, spiny dogfish, horn shark, and white-spotted bamboo shark, supplementing the sdAbs repertoires. A nanobody possesses the characteristics of small molecular weight, high affinity, strong stability, good solubility, strong tissue penetration, and recognition of hidden antigen epitopes. It has attracted increased attention in disease diagnosis, immune reagents, pathogen detection, and drug development [51]. With the development of molecular biology techniques and the improvement of genetically engineered antibody preparation technology, nanobody has become a popular research field in immunoassay. Figure 3 summarizes the development of nanobody to date.

#### 3.2.2. Nanobody Targeting Fluorescent Proteins

Fluorescent proteins have changed cell biology and biochemistry by offering simple-to-use gene-encoded fluorescent protein markers. Several tight binding agents for research and drug targets have been developed through the synthesis and selection of protein scaffold libraries during parallel development [52]. An example is the development of nanobody, which is easier to select with improved stability, solubility, and yield [53]. In contrast to traditional antibodies, these small and stable nanobodies are functional in living cells. Therefore, nanobody with specific binding activity to FP is a potent tool for FP fusion, separation, and cellular engineering in various areas of biological research.
Camelidae-derived nanobodies targeting GFP

Various proteins with excellent subcellular localization characteristics have been fused to GFP, creating visual antigens to detect proteins directly in various subcellular compartments. In 2006, Rothbauer et al. screened a GFP-specific antibody fragment (cAbGFP4) by immunizing alpaca with GFP [54]. The surface plasmon resonance assay (SPR) of the interaction between cAbGFP4 and the GFP antigen showed a high affinity (K_D_ = 0.23 nM). In addition, the anti-GFP nanobody was combined with a monomeric RFP, producing the visible GFP-binding antibody employed to examine the distribution of the anti-GFP nanobody in living cells. By gel filtration, immunoblotting, and confocal microscopy assays, GFP-specific nanobody was demonstrated to be stably distributed in mammalian cells without detectable protein degradation or aggregation commonly found with single-chain variable fragments. Furthermore, Kubala et al. characterized the GFP:cAbGFP4 complex (Figure 4A) by X-ray crystallography and isothermal titration calorimetry (ITC) [55], revealing the basis for high affinity and specificity of nanobodies in protein binding. In 2020, four distinct anti-GFP nanobodies, termed A12, B9, D5, and E6, were identified using phage display [56]. Native PAGE and immunoprecipitation assays revealed that these nanobodies could bind GFP in vitro and in vivo.

Protein conformation is closely related to function and is usually controlled by regulatory factors. Axel et al. identified seven GFP-specific binders, namely GFP-binding proteins (GBPs) 1–7 [57]. Among them, GBP1 increased GFP fluorescence by 10-fold, while GBP4 induced a 5-fold decrease in GFP fluorescence; hence, these were referred to as “enhancer” and “minimizer”, respectively. Structural analyses of the GFP–nanobody complex (Figure 4B,C) revealed that the two nanobodies caused modest opposing changes in the chromophore milieu, altering its absorption characteristics [58]. Furthermore, 25 GFP-specific nanobodies (LaGs) were identified, with K_D_ values ranging from 0.5 nM to over 20 μM [59]. A bivalent nanobody (LaG-16-LaG-2) showed the highest affinity, with a K_D_ value of 36 pM. Moreover, the maximum fluorescence intensity of GFP increased by ~60% when incubated with excess LaG protein. Zhang et al. reported the crystal structure of GFPuv complexed with LaG16 (Figure 4D) at 1.67 Å resolution [60]. The binding site of LaG16 on the GFP β-barrel was located on the other side of the GFP enhancer. Hence, LaG16 and GFP-enhancer were fused with a (GGGGS)_4_ linker. The bivalent nanobody had an affinity of 0.5 nM, demonstrating the feasibility of designing ultra-high-affinity target protein binders by dimerization of 2 nanobodies binding with different epitopes.

In 2014, Twair et al. prepared sfGFP and immunized an adult one-humped camel. Seven anti-sfGFP nanobodies targeting three epitopes (NbsfGFP01, 02, 03, 04, 06, 07, and 08) were isolated [61]. Based on ELISA and immune-blotting assays, these nanobodies recognized sfGFP labeled as free or fused to growth hormone. In addition, the crystal structure of nanobody NbsfGFP02 complexed with sfGFP (Figure 4E) was established at a resolution of 2.2 Å. The affinity between NbsfGFP02 and sfGFP was determined to be 15.8 nM by biolayer interferometry (BLI). The melting temperature was 75.6 ℃ for NbsfGFP02 [62]; hence, it is a prospective GFP nanobody candidate in applications that demand harsh testing conditions.

Most nanobodies can be functionally expressed in vivo via plasmid transfection into eukaryotic cells. Hence, nanobodies are an excellent tool for identifying structural or dynamic features in living cells. In 2020, Zhou et al. proposed a novel method of expressing the constructed GFP-binding nanobodies (cAbGFP4) as in vitro transcription (IVT) mRNA, referred to as nanobody-mCherry [63]. The mRNA with untranslated regions and reverse cap analogues capped with chemically modified nucleotides and poly(A) tail was prepared in vitro and used for transfection. In contrast to the nanobody expressed using the plasmid DNA, the anti-GFP nanobody expressed using IVT mRNA was identified within 3 h of transfection and degraded within 48 h. Therefore, expressing the encoded mRNA of nanobody in living cells allows efficient delivery of the nanobody.
Shark-derived nanobodies targeting GFP.

Wei et al. demonstrated that bamboo sharks produced an effective immune response against GFP immunization, characterized by elevated lymphocyte counts and antigen-specific IgNARs [64]. In total, 7 anti-GFP nanobodies, including BsG3, 73, 80, 89, 93, 98, and 105, with an affinity of up to 0.3 nM, were isolated from immunized bamboo sharks, implying that bamboo sharks manufacture high-affinity IgNARs. In addition, the bi-paratopic VNARs with the highest affinity to GFP (20.7 pM) were constructed, and the character of anti-GFP nanobodies as intrabodies was validated in mammalian cells. These findings will speed up the research and progress of bamboo shark sdAbs to provide low-cost and easy-to-use nanobodies for the biomedical industry.
Nanobodies targeting other fluorescent proteins.

mWasabi is a bright monomeric green fluorescent protein. Li et al. successfully constructed an antibody library of 4 × 10^7^ transformants by immunizing camels with mWasabi as an antigen, and screened 3 high-affinity mWasabi-specific nanobodies, termed Nb4, Nb6, and Nb27 [65]. These nanobodies recognized mWasabi alone or when combined with programmed death 1 (PD-1). In total, 6 nanobodies (LaMs) targeting mCherry with high specificity were identified, with K_D_ values ranging from 0.18 nM to 63 nM [59]. In addition, three iRFP713-specific nanobodies (BSR1, BSR3, and BSR4) with nanomolar binding affinities were isolated [64]. These data suggest that immunization of bamboo sharks can produce high-affinity nanobodies.

Overall, the ability of nanobodies to bind cellular proteins and attract FP fusion proteins enables precise control of cellular processes and structures in living cells. This multifunctional FP-nano trap enables microscopic, biochemical, and functional analyses with a unique combination of the same protein. Nanobodies targeting FPs are summarized in Table 3.

## 4. Applications of Fluorescent Protein Nanobody

### 4.1. Applications in Protein Detection

#### 4.1.1. Detection of Intracellular Proteins

Understanding protein function requires reliable and quantifiable high-resolution protein localization. Although the use of antibodies to label target proteins has been well established in molecular biology, this technique is constrained by the size and multivalency of conventional antibodies. Given the small size of nanobodies, they can be used as tracers for intracellular imaging after ligation with fluorescent molecules, enzymes, peptides, receptors, biotin, and other drugs. For instance, nanobodies against FP were used in super-resolution microscopy imaging when tagged with organic dyes [66,67]. Ptk2 cells that continuously express tubulin-YFP were imaged using this approach, and the resolution was improved to 269 ± 37 Å, as opposed to approximately 450 Å with conventional antibodies.

Nanobody-mediated labeling offers a quick and versatile method to label almost any commonly accessible FP-derived fusion structure for sophisticated single-molecule localization microscopy (SMLM) imaging. Specifically, two-color SMLM can investigate the subcellular localization of any functional GFP and RFP fusion constructs when nanobodies against GFP and RFP are used simultaneously [68]. Thus, numerous biological problems can be promptly addressed using two-color SMLM imaging. Ariotti et al. proposed a modular approach for enzyme-based protein labeling, allowing for improved speed and sampling for analyzing subcellular protein distributions to EM-resolution [7]. By designing GBP4 directly to the modified soybean ascorbate peroxidase (APEX) tag, it was shown that APEX could be directed to any GFP-labeled protein of interest. APEX-GBP4 fusion provides notable high-resolution protein localization to the organelle subdomains and significantly shortens the time for characterizing subcellular protein distributions. Furthermore, it permits EM-resolution of GFP-labeled proteins expressed at endogenous levels.

A toolbox of FP-specific nanobody-encoding plasmids was generated and fused into functional modules. This toolbox enables the visualization and manipulation of intracellular signaling pathways in living cells, significantly expanding its uses in vivo [9]. These include fluorescent sensors for dynamic visualization of Ca^2+^, H^+^, and ATP/ADP, and oligomeric or heterodimeric modules that allow protein recruitment or isolation and recognition of membrane contact sites between organelles. In 2017, Herce et al. used cell-penetrating peptide (CPP) to transport antibodies directly into cells for immunolabeling and antigen manipulation [69]. A system that took advantage of the high affinity of cyclic arginine-rich CPP toward RNA in the nucleolus was constructed. By linking the cyclic arginine-rich CPP to the GFP nanobody, the re-localization of GFP in the nucleolus can be directly observed in cells (Figure 5A). Therefore, this visualized system can be used to track the protein’s location and to compare the efficiency of different CPPs quantitatively.

The fluorescence resonance energy transfer (FRET) technique is widely used in life science research because it enables dynamic real-time detection of signaling molecules under physiological conditions in living cells. Due to its exceptional sensitivity and specificity, time-resolved Förster resonance energy transfer (TR-FRET)-based analysis is becoming increasingly popular in biomedical research. The nanobody-based TR-FRET method allows easy quantification of fluorescent (fusion) proteins in lysates with much higher sensitivity than conventional fluorescence intensity readouts [70].

#### 4.1.2. Detection of Proteins on the Cell Surface

The bacterial surface display is a promising technology for producing cell-anchored proteins and designing whole-cell catalysts. Although various outer membrane proteins are used for surface display, no simple, universal, and high-throughput compatible methods are available to evaluate and develop surface display systems. In addition, it is challenging to distinguish between intracellular and surface-displayed proteins. Wendel et al. constructed a fluorescence-based surface display detection system by fusing GFP-nanobody to outer membrane anchors [71]. Two commonly used outer membrane proteins were chosen as anchors: outer membrane protein A (OmpA) and autotransporter (C-IgAP). Hence, two different display modules are constructed by fusing OmpA or C-IgAP with GFP-specific nanobodies, visualized by adding purified GFP externally. Although GFP itself can be displayed on the cell surface, this new method avoids the problem of false positives since only if GFP binds to the nanobody presented on the cell surface can the cells produce a fluorescent signal. The assay is compatible with many fluorescence detection methods, including whole-cell fluorescence detection in plate, in-gel fluorescence, microscopy, and flow cytometry. This inexpensive and easy-to-read surface display method will help to demonstrate the transport mechanism of proteins onto the surface of living cells, enabling the rational development of bacterial surface display systems and robust whole-cell biocatalysts in the future.

The release of neurotransmitters requires exocytosis, endocytosis, and the formation of new fusion vesicles. What happens to vesicle proteins after exocytosis, when left on the plasma membrane, is poorly understood. These proteins are frequently conjugated to pH-sensitive GFP moieties (pHluorins). As pHluorin imaging is usually limited by the diffraction of spots several times larger than vesicles, using anti-GFP nanobodies to selectively label exocytosed vesicles is valuable [72]. By linking anti-GFP nanobodies to chemical fluorophores suitable for super-resolution imaging, the size and intensity of pHluorin-labeled proteins under various conditions can be detected in ways not possible with pHluorin alone. Upon stimulation of exocytosis, new vesicle proteins are exposed to the plasma membrane, and then the fluorescently labeled nanobodies will bind with pHluorin (Figure 5B). Thus, nanobody-based pHluorin detection is a promising tool for studying post-exocytosis events in neurons.

### 4.2. Applications in Targeted Degradation of Protein Molecules

Compared to gene editing and RNA interference, direct manipulation of biomolecules at the protein level is a more efficient route for protein function studies, overcoming limitations such as potential off-target effects, gene inactivation, and loss of essential gene phenotypic function. Targeted protein degradation is currently a major research strategy to achieve the loss of function and proteolysis of proteins of interest. A nanobody-based protein degrader can help achieve rapid degradation and reversible regulation of proteins of interest. Caussinus et al. developed a protein degradation method for the direct and rapid depletion of target GFP fusion proteins in any eukaryotic system [73]. Briefly, to knock down the GFP fusion protein, an anti-GFP nanobody is fused to the F-box protein (FBP) in the SKP1-CUL1-F-box protein (SCF) E3 ligase complexes to recognize the GFP fusion protein. The ubiquitin-conjugating enzyme (E2) covalently links multiple ubiquitin molecules to the target GFP fusion protein. Subsequently, the SCF complex degrades the polyubiquitinated protein; thus, removing the target protein is easy to monitor (Figure 5C). This technique, termed degrade green fluorescent protein (deGradFP), has been used for the degradation of GFP and its fusions in mammalian cells, zebrafish embryos [74], and plants [75].

### 4.3. Applications in Bimolecular Complementation Affinity Purification System

In 2016, Croucher et al. developed a bimolecular complementation affinity purification system (BiCAP) which can effectively distinguish human epidermal growth factor receptor (HER) dimer (homologous and heterologous) from monomer [76]. In this system, two complementary fragments of an FP molecule are fused with two HER proteins (HER1, HER2, or HER3). These two fragments cannot be spontaneously assembled into active fluorescent proteins. However, suppose the two HER proteins interact with each other. In that case, the two fragments will be spatially close to each other and complementary, reconstructing them into a complete and active fluorescent protein (Figure 5D). The HER dimer and its interacting proteins can be enriched using a GFP nanobody, which had no affinity for the GFP fragments, thus enabling the isolation and enrichment of the dimer. By analyzing the identified proteins, it was revealed that the three dimers (HER2:HER2, HER2:HER3, and HER1:HER2) have common interacting proteins and their specific interactome. FAM59A, a protein that specifically interacted with HER1:HER3 dimer, mediated the activation of the extracellular signal-regulated kinase pathway, providing a new target for breast cancer therapy.

### 4.4. Applications in Other Areas

Early in vitro diagnosis of disease and monitoring of therapeutic effects is a major problem in disease diagnosis. An ideal contrast agent should have good tissue penetration, high antigen affinity, and rapid clearance capability with minimal damage to normal tissues. Nanobodies have the potential as ideal imaging agents that can cross blood vessels and enter tissues for better imaging and therapeutic applications.

FPs not only illuminate cells and biological processes but also make excellent scaffolds due to their apparent lack of linkage to numerous host protein networks. Using GBPs and GFP as a scaffold to drive biologically active complex formation, Tang et al. created a library of hybrid transcription factors that exclusively control gene expression in the presence of GFP and its derivatives [77]. The production of GFP controls the expression of cell-specific genes (Figure 5E) and promotes the dysfunction of the mouse retina and brain. In addition, the GFP transgenic mice and zebrafish strains are modified to achieve GFP-dependent transcription for the photogenetic monitoring of neural circuits. This work establishes the position of GFP as a versatile scaffold and opens the door to selectively manipulating various GFP-tagged cells in transgenic strains. Based on this, other intracellular products can also be developed as cell-specific scaffolds in multicellular organisms.

DNA nanostructures have become an essential and effective tool for studying enzyme activity and protein function. However, developing universal strategies for forming protein complexes on DNA nanostructures is difficult. One of the difficulties is the attachment of proteins of interest to DNA nanostructures. Sommese et al. proposed a novel approach to labeling DNA nanostructures [78]. By functionalizing them with a GFP nanobody, the ability of protein attachment can be precisely controlled (Figure 5F). Compared with GFP-specific DNA aptamers, nanobodies exhibit higher specificity, stability, and affinity toward GFP. Therefore, the application of DNA nanostructures as a programmable scaffold in biological research has been dramatically simplified by connecting the DNA nanostructures with FPs commonly found in cells, developmental biology, and protein biochemistry.

## 5. Discussion on Future Outlook

Compared with traditional antibodies, nanobodies have better physical and chemical properties and are easier to express and screen. Nanobodies are much simpler in structure than conventional antibodies. They are encoded by a single gene and can be easily produced by microorganisms, significantly reducing the production cost. Although nanobodies provide a breakthrough for antibody research, some problems still need to be solved. On the one hand, they possess the disadvantage of inconvenient operation and high cost of animal immunization. Furthermore, as HCAb and IgNAR are abundant in peripheral blood monocytes, nanobodies targeting FP antigens are usually screened from specific Camelidae or shark immune libraries. With the maturation of library construction and nanobody development technologies, screening FP-specific nanobodies from synthetic libraries is a new direction which will overcome the inconvenience of immunizing animals, shorten the experimental period, and significantly reduce the cost. On the other hand, although the screening and expression of nanobodies is relatively simple, obtaining FP nanobodies with potential application values is a very complicated process. The screening of phage nanobody libraries cannot avoid the false positives caused by the binding of filamentous phage surface with antigen, while the small capacity of yeast and bacterial nanobody libraries is another difficult problem in nanobody screening.

The applications of nanobodies against FP will continue to increase in the future. (1) With the development of intracellular antibody technology, detecting intracellular molecules in living cells has become accessible. Nanobodies have unique advantages in this respect, in that (i) the nanobody is small, and the efficiency of cell entry is much higher than that of traditional antibodies; and (ii) the nanobody can be expressed and functional in living cells. (2) Bispecific nanobody is a kind of artificially modified antibody that can specifically bind two different antigens simultaneously. It can be easily constructed from monovalent nanobodies and expressed in microorganisms. Hence, nanobodies against FP provide a broad application prospect for bispecific nanobody development.

## 6. Conclusions

This paper reviews the origin, structure, and properties of FPs. Monoclonal antibodies and nanobodies targeting FPs and their applications are also summarized. Although antibodies are valuable tools for displaying biological components in immobilized cells, the use of traditional antibodies in living cells is constrained by the ineffective folding and assembly of their variable heavy and light chains. Direct microinjection of antibodies is the primary method used in antibody intracellular applications, which is technically challenging and stressful to cells. Camel or shark-derived single-domain antibodies recognize antigens through their variable domains of heavy chains. These small and stable nanobodies can be expressed and functional in living cells. Therefore, generating nanobodies targeting FPs will make FPs more valuable in biological research.

## Figures and Tables

**Figure 1 ijms-24-04307-f001:**
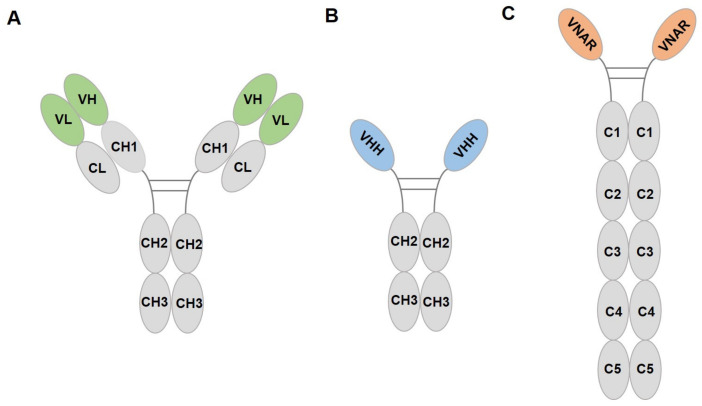
Structure diagrams of IgG (**A**), camelid HCAb (**B**), and shark IgNAR (**C**). VH, variable region of heavy chain; VL, variable region of light chain; CH/C, constant region of heavy chain; CL, constant region of light chain; VHH, variable domain of HCAb; VNAR, variable domain of IgNAR.

**Figure 2 ijms-24-04307-f002:**
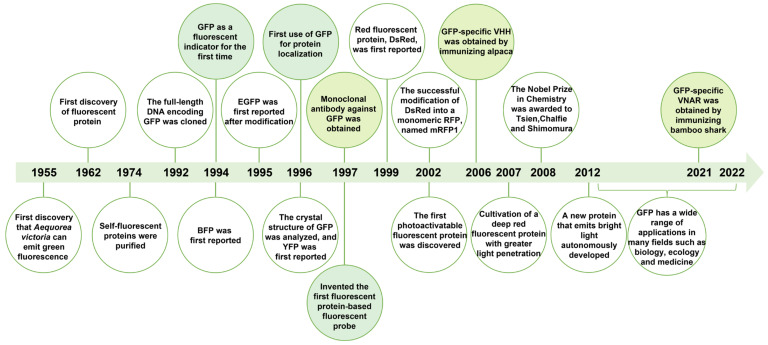
The development of fluorescent proteins. GFP, green fluorescent protein; BFP, blue fluorescent protein; EGFP, enhanced green fluorescent protein; YFP, yellow fluorescent protein; RFP, red fluorescent protein; VHH, variable domain of heavy chain; VNAR, variable new antigen receptor. The pale green marker indicates the progress of research on GFP application and the yellow green marker indicates the progress of research on GFP antibody.

**Figure 3 ijms-24-04307-f003:**
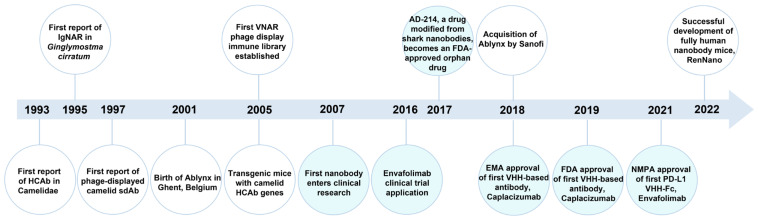
The development of nanobody. The light blue marker indicates the progress of research on nanobody drugs. FDA, Food and Drug Administration; EMA, European Medicines Agency; NMPA, National Medical Products Administration; HCAb, heavy-chain antibody; IgNAR, Ig new antigen receptors; sdAb, single domain antibody; VHH, variable domain of HCAb; VNAR, variable domain of IgNAR; PD-L1, programmed cell death 1 ligand 1.

**Figure 4 ijms-24-04307-f004:**
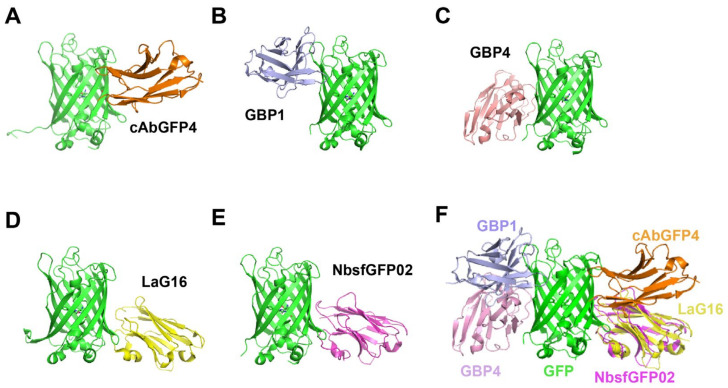
Diagram of GFP–nanobody complex. (**A**) cAbGFP4 (PDB:3OGO, orange); (**B**) GBP1 (PDB: 3K1K, light blue); (**C**) GBP4 (PDB: 3G9A, pink); (**D**) LaG16 (PDB: 6LR7, yellow); (**E**) NbsGFP02 (PDB: 7E53, magenta). (**F**) Structural superposition of above nanobodies with GFP complex. GFP is shown in green color.

**Figure 5 ijms-24-04307-f005:**
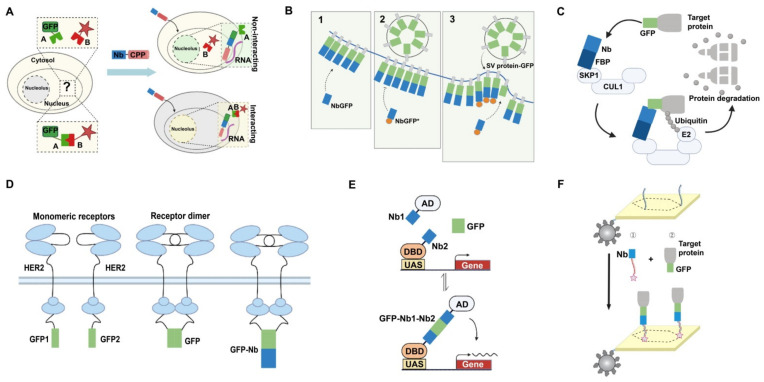
Applications of fluorescent protein nanobody. (**A**) Visualization of protein–protein interactions. Anti-GFP nanobody (Nb) bound to cell-penetrating peptides (CPPs) can penetrate the cell nucleus. If the GFP-tagged protein does not interact with the RFP-tagged protein, only the GFP-tagged protein re-localizes to the nucleolus via Nb-CPP, showing green color. However, if the two proteins interact, both proteins re-localize to the nucleolus, showing yellow color. (**B**) GFP nanobody enables the selective labeling of newly exocytosed vesicle proteins. Step 1: The vesicle protein is coupled to GFP and incubated with a non-fluorescent nanobody (NbGFP), resulting in the inaccessibility of GFP with the fluorescence-binding nanobody (NbGFP*). Step 2 and 3: After stimulation of cellular exocytosis, the new vesicle protein is exposed to the plasma membrane and can be labeled by NbGFP*. (**C**) Schematic illustration of deGradFP. Protein selective degradation is accomplished through the ubiquitin pathway, which is carried out by a complex series of enzymes. The F-box protein (FBP) in the SKP1-CUL1-F-box protein ligase complexes (SCFs) is fused to the anti-GFP nanobody (Nb) to recognize the GFP fusion protein, and the ubiquitin-conjugating enzyme (E2) covalently links multiple ubiquitin molecules to the target protein. Subsequently, the polyubiquitinated protein is degraded by the proteasome. (**D**) Schematic description of BiCAP. HER protein is fused to N-terminal (GFP1) or C-terminal (GFP2) fragments of GFP. When the two fusions form a dimer, GFP refolds and fluoresces. The HER dimer and its interacting proteins are then enriched using a GFP nanobody, which has no binding ability toward the GFP fragments and, thus, enables the isolation of the dimer. (**E**) GFP-dependent transcription system. The DNA-binding domain (DBD) and the activation domain (AD) are fused with GFP nanobody 1 (Nb1) and GFP nanobody 2 (Nb2), respectively. In the presence of GFP, DBD-Nb1 and AD-Nb2 can activate the upstream activating sequence (UAS) reporter gene, which leads to the upregulation of target genes. (**F**) Recognition of GFP-labeled proteins on DNA nanostructures. The DNA nanostructures with two binding sites are ligated onto the magnetic resin, bound with GFP nanobody (Nb) by complementary base pairing, and then incubated with GFP-labeled proteins. Hence, GFP-labeled proteins can be accurately attached to the nanostructures.

**Table 1 ijms-24-04307-t001:** Photophysical properties of GFP, RFP, and their derivatives.

Classification	Protein	Excitation Peak (nm)	Emission Peak (nm)	Molar Extinction Coefficient (×10^3^ M^−1^ cm^−1^)	Quantum Yield	Brightness (Relative to EGFP, %)	Structure
Green	wtGFP	395	509	25	0.79	48	monomer
	EGFP	488	507	56	0.60	100	monomer
	sfGFP	485	510	83.3	0.65	160	monomer
	GFPuv	395	509	N.A.	N.A.	N.A.	monomer
Blue	EBFP	380	440	29	0.30	27	monomer
	Azurite	383	447	26.2	0.55	43	monomer
	EBFP2	383	448	32	0.56	53	monomer
	mTagBFP	402	457	52	0.63	98	monomer
Ultramarine	Sirius	355	424	15	0.24	11	monomer
Cyan	ECFP	433	475	26	0.40	39	monomer
	Cerulean	433	475	32.5	0.6	79	monomer
	mTFP	462	492	64	0.85	162	monomer
	mTurquoise	434	474	30	0.84	75	monomer
Yellow	EYFP	514	527	83.4	0.60	150	monomer
	mCitrine	516	529	77	0.8	174	monomer
	mVenus	515	528	92.2	0.60	156	monomer
Red	DsRed	558	583	75	0.79	165	tetramer
	TagRFP	555	584	100	0.48	143	monomer
	mCherry	587	610	72	0.2	43	monomer
	mStrawberry	574	596	90	0.29	78	monomer
	dTomato	554	581	69	0.69	142	dimer
	mTangerine	568	585	38	0.30	34	monomer
	mKate	588	635	45	0.33	45	monomer
	mKate2	588	635	62.5	0.4	74	monomer
	Katushka	588	635	65	0.34	67	dimer
Orange	mOrange	548	562	71	0.70	148	monomer
	mBanana	540	553	6	0.70	12.5	monomer

N.A.: not available.

**Table 2 ijms-24-04307-t002:** Photophysical properties of NIR FPs.

Photoreceptor	Protein	Excitation Peak (nm)	Emission Peak (nm)	Molar Extinction Coefficient (×10^3^ M^−1^ cm^−1^)	Quantum Yield	Brightness (Relative to miRFP670nano, %)
BphP	IFP1.4	684	708	88	7	N.A.
	IFP2.0	690	711	86	8	N.A.
	mIFP	683	704	82	8.4	67
	miRFP670	642	670	87.4	14	119
	miRFP703	674	703	90.9	8.6	76
	miRFP709	683	709	78.4	5.4	41
ApcF	BDFP1.5	688	711	74	5.0	36
CBCR	miRFP670nano	645	670	95	10.8	100
	miRFP670nano3	645	670	129	18.5	233

N.A.: not available.

**Table 3 ijms-24-04307-t003:** Summary of nanobodies targeting FPs.

Nanobodies	Targets	Source	Reference
cAbGFP4	GFP	alpaca	[54]
GBP1-7	GFP	camel	[57]
LaGs	GFP	llama	[59]
NbsfGFP01, NbsfGFP02, NbsfGFP03, NbsfGFP04, NbsfGFP06, NbsfGFP07, NbsfGFP08	sfGFP	camel	[61]
D5, E6, A12, B9	GFP	alpaca	[56]
BsG3, 73, 80, 89, 93, 98, 105	GFP	bamboo shark	[64]
Nb4, Nb6, Nb27	mWasabi	camel	[65]
LaMs	mCherry	llama	[59]
BSR1, BSR3, BSR4	iRFP713	bamboo shark	[64]

## Data Availability

Not applicable.

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
