# Peer review of "Research Progresses and Applications of Fluorescent Protein Antibodies: A Review Focusing on Nanobodies"

_ijms, 2023, doi:10.3390/ijms24054307_

Round 1

Reviewer 1 Report

Dear Author, the review article "Research progresses and applications of fluorescent protein antibodies: A review focusing on nanobodies" focused on a novel concept for the use of fluorescent proteins (FPs) in biological research has been encouraged by the discovery of these proteins, their rich fluorescence spectra, and their photochemical properties and very well described the development of fluorescent proteins and photo-physical properties of GFP, RFP and their derivatives part; but some minor correction need to rewrite the abstract, revised the part of applications of fluorescent protein nanobody and add commercialization of FPs in order to more clarify and better highlight the findings of this review. The manuscript appears to fit the journal's aims and is appropriate. Rectify all grammatical errors. Spell out all abbreviations in the text if it first time mentioned in the text. Cross-reference all of the citations in the text with the references in the reference section. Make sure that all references have a corresponding citation within the text and vice versa. 

Author Response

Q1: The review article "Research progresses and applications of fluorescent protein antibodies: A review focusing on nanobodies" focused on a novel concept for the use of fluorescent proteins (FPs) in biological research has been encouraged by the discovery of these proteins, their rich fluorescence spectra, and their photochemical properties and very well described the development of fluorescent proteins and photo-physical properties of GFP, RFP and their derivatives part; but some minor correction need to rewrite the abstract, revised the part of applications of fluorescent protein nanobody and add commercialization of FPs in order to more clarify and better highlight the findings of this review. The manuscript appears to fit the journal's aims and is appropriate. Rectify all grammatical errors. Spell out all abbreviations in the text if it first time mentioned in the text. Cross-reference all of the citations in the text with the references in the reference section. Make sure that all references have a corresponding citation within the text and vice versa. 

A1: We thank the reviewer for raising these issues. We have rewritten the abstract (Abstract), revised the part of applications of fluorescent protein nanobody (section 4) and added commercialization of FPs (section 2). We apologize for some grammatical errors and other issues, and we have carefully revised our manuscript. 

Reviewer 2 Report

The review from Chen et al. reports on the application of nanobodies for binding to fluorescent proteins. After a broad description of fluorescent proteins, the review focuses on nanobodies, stating that they are more convenient with respect to antibodies, considering their easiness in expression and improved chemical and physical properties. The review is nicely written and presents many recent references. However, some points deserve to be better explained, as follows.

1.     In the introduction, I would recommend to be more clear in the differentiation between antibody and nanobody, providing a rigorous definition for the latter (i.e. for the nanobody).

2.     Please add figure in the introduction so that the differences between antibody and nanobody are clear.

3.     In section 3, add a brief description of monoclonal antibody, stating why it is so important in biosensing, in comparison to polyclonal antibody.

4.     The section dealing with applications is nicely written.

5.     What is the advantage of nanobodies with respect to DNA aptamers for binding to fluroscent proteins?

6.     Please also report, if possible this recent article dealing with time resolved FRET. DOI: 10.1002/anse.202200020

Author Response

The review from Chen et al. reports on the application of nanobodies for binding to fluorescent proteins. After a broad description of fluorescent proteins, the review focuses on nanobodies, stating that they are more convenient with respect to antibodies, considering their easiness in expression and improved chemical and physical properties. The review is nicely written and presents many recent references. However, some points deserve to be better explained, as follows.

Q1: In the introduction, I would recommend to be more clear in the differentiation between antibody and nanobody, providing a rigorous definition for the latter (i.e. for the nanobody).

A1: We thank the reviewer for raising this issue. We have added a description of the nanobody in the introduction section.

Q2: Please add figure in the introduction so that the differences between antibody and nanobody are clear.

A2: Thanks for your suggestion. Structure diagrams of antibody and nanobody (Figure 1) are added in the introduction section to highlight the difference between them.

Q3: In section 3, add a brief description of monoclonal antibody, stating why it is so important in biosensing, in comparison to polyclonal antibody.

A3: Thanks for reminding. We have added a brief description of monoclonal antibody in section 3 and stated the importance of monoclonal antibody in biosensing and bioimaging in comparison to polyclonal antibody.

Q4: The section dealing with applications is nicely written.

A4: Thank you very much.

Q5: What is the advantage of nanobodies with respect to DNA aptamers for binding to fluorescent proteins?

A5: Compared with GFP-specific DNA aptamers, nanobodies exhibit higher specificity, stability, and affinity toward GFP. Relevant description has been added in section 4.4.

Q6: Please also report, if possible this recent article dealing with time resolved FRET. DOI: 10.1002/anse.202200020

A6: As suggested, we have added the content of this article in section 4.1.1 (ref 70).

Reviewer 3 Report

This review comprehensively introduced fluorescent proteins and nanobodies and summarized the application of nanobody-fluorescent protein pairs. Overall, the whole manuscript is well structured and reads well, I don’t have any comments or questions, and it can be accepted in the present form.

Author Response

Q1: This review comprehensively introduced fluorescent proteins and nanobodies and summarized the application of nanobody-fluorescent protein pairs. Overall, the whole manuscript is well structured and reads well, I don’t have any comments or questions, and it can be accepted in the present form.

A1: Thank you very much.